# Inhabited Institutionalism

**Callie Cleckner *** and **Tim Hallett**

Department of Sociology, Indiana University Bloomington, Bloomington, IN 47405, USA
* Correspondence: ccleckn@iu.edu

**Definition:** Inhabited Institutionalism is a meso-level theoretical approach for evaluating the recursive relationships among institutions, social interactions, and organizations. This theoretical framework offers organizational scholars a multi-faceted consideration of coupling configurations that highlight how institutional processes are maintained, challenged, and transformed without reverting to nested yet binary arguments about individual agency and structural conditions.

**Keywords:** sociological theory; social interaction; organizations

## 1. Development of Inhabited Institutionalism

Sociologists, especially new institutionalists, lack a consistent definition of what an "institution" is; rather, the tradition often relies on vague conceptualizations [1]. Nevertheless, organizational sociologists commonly treat institutions as "broad structures of meaning that are taken-for-granted and organize activity" [2], p. 214. Similarly, Barley defines institutions as "social forms or templates composted of clusters of conventions that script behavior to varying degrees in given contexts" [3], p. 495. This conceptualization is perhaps similar to others developed across organizational studies and economics [4,5]. However, despite parallel definitions and plenty of literature, much new institutional scholarship treats institutions as "reified abstractions" void of social interaction and meaning-making [2]. Instead of attempting to redefine institutions as the object of analysis, inhabited institutionalism employs the concept of "institutional myth" to refocus empirical analyses.

Inhabited institutionalism complements the macro, anti-individualistic focus of the long-dominant New Institutionalism (NI), which Meyer and Rowan advanced sociologically, most notably via formal structures as "myth and ceremony" [6]. In their groundbreaking article, Meyer and Rowan define institutional *myths* as the prevailing practices and procedures organizations adopt to maintain legitimacy, stability, and resource systems in a larger field [6]. In the macro sense, mythologies are the rationalized and impersonal ways in which we institutionalize rules, authority, and bureaucracy. At the local level, mythologies are the ways in which organizations ceremonially present a tight linkage between formal structures and practical activity [6–8]. Institutional myths are mythic in the anthropological sense because they are cultural explanations for how the world operates, but they are also occasionally mythic in the sense that organizational operations are inconsistent with ideal expectations [8]. New institutionalists refer to this façade of commitment as "ceremony"—it can be simpler and even more effective to commit to the myth verbally instead of via genuine practices. Examples of popular institutional myths are "professionalism", "expertise", or "accountability," which vary in meaning by field and vary in adherence by organization.

Scholars have since applied myth and ceremony to observe how institutions tend to progressively homogenize because embedded organizations will at least ceremonially adhere to prominent institutional myths to maintain external credibility, a process termed "institutional isomorphism" [9]. In other words, the success of organizations depends on how effectively they can secure legitimacy and resources, which means following status quo expectations, even if members do so only in name rather than practice for the sake of

true efficiency. For instance, many companies claim trustworthiness for self-promotion, but one can imagine how attempting to improve profits may open avenues for distrustful practices. Thus, even if institutions appear similar due to explicitly adopting the same popular mythologies, on the ground practices vary. Nevertheless, adhering to dominant mythologies in the field enables organizations to appear similar across institutions, despite contextual differences. This process explains why most American universities promote campus diversity or why businesses adopted more sanitation protocols during the COVID-19 pandemic—adherence should confer more legitimacy.

Incorporating rational mythologies should allow organizations to be more legitimate, successful, and enduring—if participants are committed to maintaining the ceremonial structure because they are content with the technical manifestations [6]. Meaning, if a mythology is well-received by members because it is well-implemented, the organization should prosper. However, genuine implementation of macro mythologies can also hinder practical activity and stimulate conflict, which is why scholars have taken a particular interest in processes of decoupling or loose couplings [8,10].

The ways in which organizations legitimate mythologies by linking formal structures with practices is often defined as a process of coupling, with researchers describing activities as "decoupled" or "loosely coupled" to mythologies when organizations are only ceremonially committed to change [2,10,11]. Loose couplings may help maintain an institution's myth and ceremony, and therefore provide legitimacy. For example, many organizations now purport diversity policies and programming without a sincere coupling to systematic or cultural change [12]. Take for instance former State Farm employees and customers accusing the organization of racial discrimination, despite the company denying this reflects internal culture [13]. It is simple (and legitimatizing) to *claim* your company is not racist but *being* anti-racist with insurance claims is not profitable. Scholars have documented similar forms of decoupling or loose coupling in housing and hiring practices—organizations claim race-neutrality, yet their outcomes remain unequal [12].

Alternatively, if recoupling occurs in which "myths become incarnate" (or go from loosely to tightly coupled), environments may destabilize and experience conflict [7]. Consequently, tight couplings may stimulate uncertainty and conflict, which can lead to organizational disruptions and even threaten institutional legitimacy if members perceive imposed practices to be ineffective or insincere [7]. For example, if an organization genuinely attempts to ameliorate inequalities via diversity policies, they may receive backlash from pre-existing dominant members who perceive them as unfair [14]. On the other hand, an attempted recoupling to diversity and equity can also instigate conflict if historically underrepresented members find the practices unsatisfactory for opposing reasons.

Scully and Creed first used the term "inhabited institution" during a conference presentation in 1997, as they argued that people "not only inhabit this process [of institutional diffusion], but they actually reshape (and are reshaped by) the objects and dynamics of diffusion" [15]. Therefore, inhabited institutionalism developed through a dialogue between NI and forms of interactionist social psychology. The goal is to understand how the macro, extra-local institutional pressures that bear on organizations are inhabited by "people doing things together", whether in concert or conflict [2,16]. Organizations and institutions cannot exist without individuals propelling them in some capacity. Inhabited Institutionalism considers the dynamics among social interactions, institutions, and organizations to examine how society operates, without reducing our analysis to individual agents or omnipresent structures. Individuals do not create change on their own, and structures only exist because we instill them with meaning. Thus, the interactions in the middle are where push comes to shove.

## 2. New Institutionalism, Agency, and Social Interaction as the Solution

Although new institutionalism was crucial to understanding how myth and ceremony have contributed to isomorphism, some scholars took issue with the lack of individual agency [8]. DiMaggio (1988) criticized NI for subverting human agency, introducing the

concept of "institutional entrepreneurs" [17], whereas others used "institutional work" to tackle this concern [8]. Nevertheless, this conceptualization of the agentic institutional entrepreneur or institutional work more generally still creates a binary rather than a holistic view of human behavior. Individual actors become the bottom line—the case is either one of personal agency or cultural dopes [8]. Rather, social interactions should be the object of our analysis.

Moreover, institutional theorists have relied on a "nested" approach: conceptualizing individuals as nested inside organizations which are nested inside institutions (Figure 1) [8]. However, inhabited institutionalism proposes that the full products of social interactions are more important than the individual ingredients [7]. How people act in conjunction with their environment is far more empirically valuable than pinpointing the effects of the environment or individual agency in isolation.

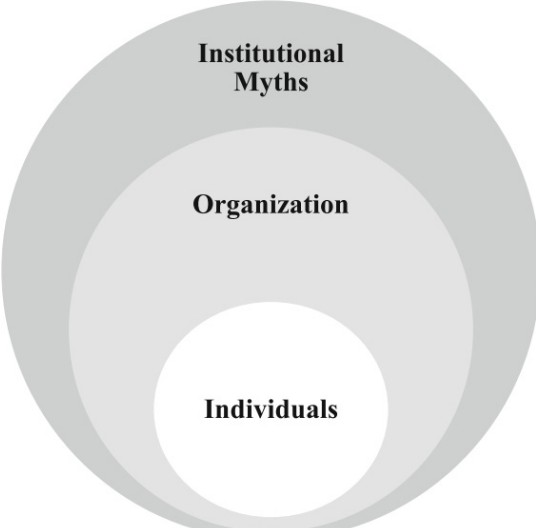

**Figure 1.** Nested theoretical approach, adapted [8].

*2.1. Including Social Interaction*

Institutions are inhabited by individual actors, but it is among their social interactions with others that we find organizational variety. Further, inhabited institutionalism does not require us to examine social interactions in isolation; rather, they are examined in relation to institutional mythologies and organizational constraints [8]. Adapting Blumer's three famous premises, inhabited institutionalism presupposes that people in organizations (1) act toward institutional mythologies based on the meanings they have to them. (2) The meanings of mythologies are developed via social interaction, and (3) these meanings are modified through an interpretive process [18]. Inhabited Institutionalism expects the people within organizations to have varied understandings of myths rather than assuming formal structures are homogenously implemented and understood across organizations. Moreover, the expectations of social interaction create additional constraints on how individuals should behave, but they can also create semi-autonomous spaces of opportunity that might not be visible in isolation [19,20]. These spaces are where "agency" may flourish despite organizational and institutional forces.

Thus, social interactions are best understood in relation to local organizational constraints and broader institutional mythologies. Focusing on this triumvirate of social interactions, institutional mythologies, and the organization, inhabited institutionalism considers the full range of "couplings" that comprise an organization. NI primarily considers the loose coupling between institutional mythologies and organizational practices (i.e., myth and ceremony), whereas scholars in the inhabited tradition document a range of coupling configurations including re-couplings (a change from loose to tight), tight couplings, loose couplings, and complex combinations [7,21–24]. Across empirical cases,

inhabited institutionalism seeks to understand the variety of coupling configurations in play because they provide insights to the distribution of opportunities and resources as well as the social construction of reality more generally [25].

### 2.2. Coupling Configurations

Coupling configurations are a contrast to the "nested" approaches in dominant institutional theory, as represented in the adapted Figure 1 [8]. Social life is more multi-dimensional than this imagery suggests. Figure 2 depicts a more comprehensive approach through the imagery of linked spheres or "coupling configurations" developed in inhabited institutionalism [8]. Figure 2 represents a fairly loosely coupled configuration, the model presumed by NI, with the important inclusion of social interaction [8]. Again, this is because individuals cannot create change with purely their own volition—change is an inherently interactive process. Returning to a previous example, most universities subscribe to the myth of diversity in some capacity. However, this commitment is often only loosely coupled to social interactions and campus organizations at predominately white institutions [12,26]. In fact, some research suggests white students benefit the most from "diverse" interactions in higher education compared to other races/ethnicities [27,28]. Despite limited meaningful engagement, there are times when the diversity myth is tightly coupled to one's social interactions—say for instance, someone asks a black student "are you on a diversity scholarship?" The coupling between social interaction and the institutional myth dramatically tightens. The way this hypothetical university implements "diversity and equity" might be loosely coupled to this student's experience in general, but occasionally it becomes tightly coupled in their interactions with others who bring the myth to the foreground.

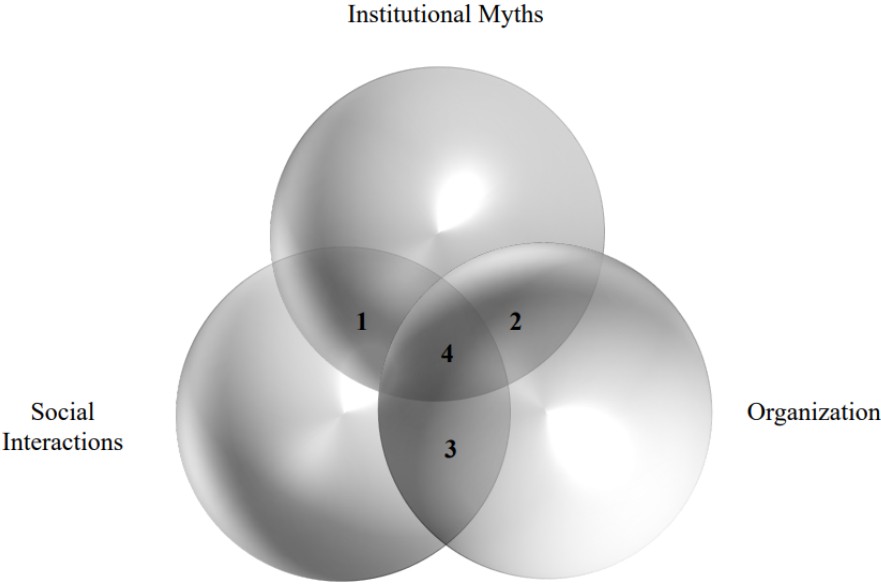

1: The coupling between institutional myths and interaction
2: The coupling between the organization and institutional myths
3: The coupling between interaction and the organization
4: The coupling between institutional myths, interaction, and the organization

**Figure 2.** Depicts the linked spheres (i.e., coupling configurations), adapted [8].

Therefore, it is important to note coupling configurations can be complex and quickly changing. Nevertheless, this is the primary strength of the framework—scholars can highlight moments and examples that contribute to much larger social processes without

reverting to the agency versus structure dichotomy. The following subsections detail the differences between tight and loose coupling configurations as well as examples of mixed situations.

### 2.2.1. Loose Coupling Configurations

The purpose of arranging these three levels of analysis in this format is to show how the factors can be separate yet still exert varied force on one another. The less "force" or authority an organization or institution has over social interactions, the more loosely coupled the configuration is. For instance, No Child Left Behind was introduced to improve testing outcomes in the US, yet in turn, the policy exacerbated inequalities in education rather than providing a safety net as the name (i.e., myth) suggests [29]. A looser coupling to the accountability myth means a school has greater flexibility in maintaining standardized outcomes, perhaps because it is a state with easier than average tests or excessive cheating [30,31]. If the state administers extremely easy exams, the accountability myth is loosely coupled to organizations (K-12 schools) and social interactions (teaching) because it is not a major concern to the individuals in this context. The accountability myth does not have a strong influence on how educators, administrators, and students must behave. This example could be represented by Figure 2.

### 2.2.2. Tight Coupling Configurations

Tight coupling configurations occur when an institutional myth, organizational practice, or social interaction are strongly influenced by one another. Figure 3a represents a slightly tighter configuration, as compared to Figure 2, whereas Figure 3b represents a very tightly coupled configuration. Tight coupling configurations mean two (or more) entities have a meaningful relationship between them in a situation. For instance, in American higher education, SAT scores were once believed to be the best determinant for college admissions and future academic success, despite SAT scores being poor predictors of student outcomes and being culturally biased [32]. The SAT myth was tightly coupled to how universities and colleges operated for decades and in turn, how students behaved, because whether or not one believed SAT scores predicted academic propensity, we treated it as such. Now that colleges and universities have begun to move away from requiring standardized test scores, we can conceptualize a university with a test-optional policy like Figure 3a—standardized test scores can still affect whether you gain admission to a university, but there is more opportunity space compared to a university that nearly requires high standardized test scores for admission (Figure 3b).

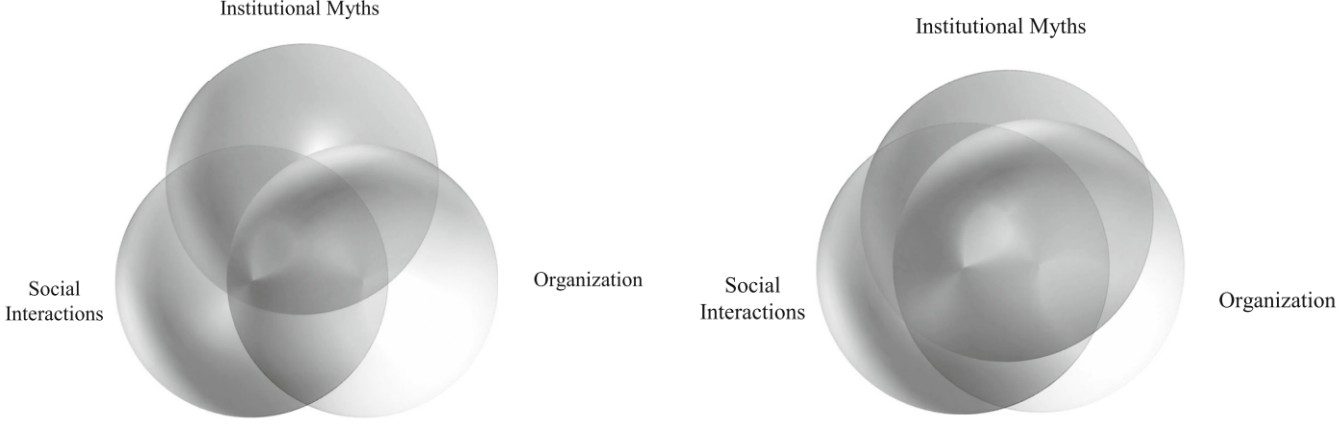

(**a**) A more tightly coupled configuration            (**b**) Very tightly coupled configuration

**Figure 3.** (**a**) represents a tighter coupling configuration compared to Figure 2, and (**b**) is an even more tightly coupled configuration, adapted [8].

### 2.2.3. Complicating Coupling Configurations

Coupling configurations are not always entirely loose or tight. In other words, the relationships between two entities might be tight, while they are loosely connected to the third and vice versa (two loose, one tight). Organizations have policies, rules, and regulations that are created to perpetuate institutional myths such as professionalism, meritocracy, or colorblindness [33]. However, how members of the organization behave is not always in accordance with organizational constraints or cultural ideals. For instance, food workers have strict guidelines for kitchen safety and sanitation. Most consumers hope restaurants are always tightly connected to this ideal in theory and practice (Figure 3b). Regardless, there are plenty of instances where social interactions are loosely coupled to organizational and institutional commitments to safety and sanitation. Take for instance a restaurant employee telling a new line cook he does not have to wear gloves and a hairnet if there is a rush. Now, the coupling configuration looks like Figure 4 in which the sanitation rules are still tightly coupled to the myth, yet the social interactions are only loosely aligned. There is more flexibility in the way the restaurant employees can behave in this type of configuration.

By incorporating an interactionist approach, inhabited institutional scholars can show the complex, situational yet constantly connected factors in any given scenario. This offers endless conceptual models across domains of social life. While this may appear daunting, these models point to how broader social processes operate. The puzzle pieces (i.e., data) must be found and organized before we can put together the full picture. Then, the most prominent and apt examples can provide a snapshot of how the much broader phenomena of interest function.

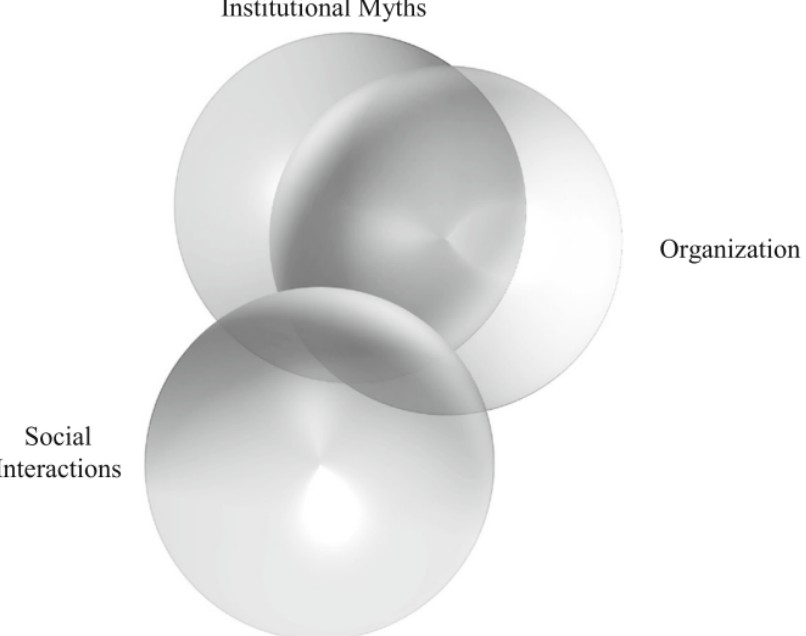

**Figure 4.** Institutional and organizational tight coupling, loosely coupled to social interactions, adapted [8].

It is also important to note that tight and loose couplings themselves are value neutral. There is no "correct" way for a coupling configuration to exist in the abstract because it is context dependent. There are instances where an organization might want to be loosely coupled or even decoupled from a myth, say something sinister like white supremacy or eugenics, and there are instances where they might attempt to be tightly coupled to a myth like diversity and inclusion. On the other hand, an organization may be tightly coupled to the myth of white supremacy, take the extreme example of Neo-Nazis. Members are content with a tight coupling between their organization, white supremacy, and the

way they interact with and treat others. Thus, tight versus loose linkages are value free descriptors of social life. Whether a coupling *should* be loose or tight is not for sociologists to decide. We are simply evaluating the magnitude to which it exists empirically. Whether that is the way things *ought* to be is up to readers and those operating and experiencing the forces we discuss.

Critics may question the utility of this perspective because it may seem vague, abstract, or empirically limited. However, if we use the perspective for a comparative analysis, the benefits become clear. For instance, scholars are interested in how American teachers approach education differently despite similar training [7,23,33]. If a school is tightly coupled to a myth such as colorblindness, teachers are going to have a more difficult time teaching students in a race-conscious manner [33]. Whereas if another nearby (but perhaps less well-funded) school is loosely coupled to the same myth due to different constraints, teachers have more flexibility and freedom to teach in a way they find more equitable [33]. This shows how the local meaning and culture in an organization can influence outcomes even if it might appear systematically similar to others in the field.

### 2.3. Guidelines for an Inhabited Approach

In their foundational article, Hallett and Ventresca propose three "signposts" for the inhabited institutions approach: (1) acknowledging the embeddedness of institutions and interactions, (2) an emphasis on meanings (both locally and broadly), and (3) a skeptical, inquiring attitude [2]. These three tenets broadly guide scholars in this tradition—maintaining a critical view on how intertwining institutions, organizations, and social interaction influence social outcomes is key to both understanding social life and potentially improving it. Coupling configurations provide empirical leverage, but it is important to remember their theoretical conceptualization, which is based on combining the principles of social interactionism with the macro ideas in new institutionalism. While some may argue the perspective is a "kitchen-sink" approach, social life is a combination of multiple, changing intertwined factors with varied salience across one's experiences. Rather than forcing our arguments into one element, we should be developing ways to simultaneously and systematically evaluate each of these moving pieces. Although sociologists prefer to organize analyses by level, the micro, meso, and macro are in constant flux with one another.

### 3. The Future of Inhabited Institutionalism

Sociologists are increasingly identifying how organizations maintain and challenge axes of difference, including race/ethnicity, gender, sexuality, social class, and other positions of power and privilege [12,27,34–37]. Inhabited institutionalism offers a comprehensive framework for observing and analyzing these dynamic, multi-dimensional processes across organizations. Social life is complicated; thus, we need complex models to envision how phenomena occur across situations and contexts. Researchers should continue to adopt this framework so that we can collaboratively elucidate social constructions of race/ethnicity, gender, sexuality, and more without relying on the false dichotomy of structure versus agency.

The inhabited viewpoint remedies debates about how the world operates broadly because it simultaneously incorporates all avenues for change. It is not enough to claim "systems" or "structures" determine life chances and experiences, nor is it enough to argue everyone has equal opportunities in America based on their individual merits. We must show how it is the case that certain identities and statuses can generally be privileged over others despite success stories. Inhabited Institutionalism helps reveal this gray area that truly colors life experiences.

**Author Contributions:** Conceptualization, T.H. and C.C.; writing—original draft preparation, C.C.; writing—review and editing, T.H. All authors have read and agreed to the published version of the manuscript.

**Funding:** This research received no external funding.

**Institutional Review Board Statement:** Not applicable.

**Informed Consent Statement:** Not applicable.

**Conflicts of Interest:** The authors declare no conflict of interest.

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
