# Peer review of "Inhabited Institutionalism"

_encyclopedia, doi:10.3390/encyclopedia2030101_

Round 1

Reviewer 1 Report

This a coherent and well rounded text. I enjoyed reading it. I think it dea it's job of presenting a more sophisticated theory of neo-institutionalism by coopting social interactionism.

Author Response

Thank you for your review. 

Reviewer 2 Report

I am not the most appropriate reviewer for this article, because I am an economist, not a sociologist. I agreed to review because I was interested in reading the article. I do some work in the new institutional economics, and I see come commonalities there, but maybe some differences as well. Would it be worth making a short comparison?

Economists would define institutions along the lines of "the humanly devised constraints that structure interactions among individuals." Oliver Williamson and Douglass North (both Nobel laureates) are big names in the new institutional economics. Taking this perspective, legal institutions obviously come to mind. But businesses and schools are also institutions. You discuss schools so I mention them.

Institutional myths have not quite made it into institutional economics, although propaganda and control of information seem closely aligned to that idea. Economists tend to take a more individualistic approach to social interaction (and I think could benefit from recognizing the impact on group membership from sociologists). 

As someone who is reading outside his field, the paper gives me a general understanding of the concept of inhabited institutionalism, but after reading the article, it was unclear to me how sociologists define institutions. I gave you an economist's definition above. If you hope to have any degree of interdisciplinary readership, perhaps you should begin by defining institutions or institutionalism before explaining inhabited institutionalism.

Reviewer 3 Report

The authors define an inhabited Institutionalism as a theoretical approach for evaluating the relationships among institutions, social interactions, and organizations and describe the ways in which organizations legitimate and build mythologies and draw a variety of coupling configurations possible within the existing myth and institutional structures and their impact on everyday reality. All models of configurations are well illustrated with examples. Th authors point to importance of paying attention to a combination of multiple, changing and intertwined factors across individual' s experiences and contexts in all configurations rather than forcing on one element in the system. I suggest considering this piece of theoretical writing for inclusion in Encyclopedia.

Author Response

Thank you for your review. 

Reviewer 4 Report

The paper is well structured and written. 

Congratulations on your study - the theoretical framework is thought-provoking and valuable for scholars to develop effective research motivations and questions.

The paper shows a clear definition of the concept of inhabited institutionalism. The logical argument is very attractive. Also, the paper has its own consistency of structure which follows the readers to understand it clearly.   Further correction might hurt its advantage.

Author Response

Thank you for your review. 

This manuscript is a resubmission of an earlier submission. The following is a list of the peer review reports and author responses from that submission.